# Quantification of Serum Exosome Biomarkers Using 3D Nanoporous Gold and Spectrophotometry

**DOI:** 10.3390/s22176347

**Published:** 2022-08-24

**Authors:** Amera Al Mannai, Tareq Al-Ansari, Khaled M. Saoud

**Affiliations:** 1College of Science and Engineering, Hamad Bin Khalifa University, Doha P.O. Box 34111, Qatar; 2Liberal Arts and Science Program, Virginia Commonwealth University, Doha P.O. Box 8095, Qatar

**Keywords:** breast cancer, exosomes, porous gold, none porous gold nanoparticles, detection

## Abstract

Tumor-derived exosomes may provide biomarkers for cancer treatment. Using sputtering technology, an affinity-based device to capture exosomes was developed using nanoporous substrate (NPG)-coated silicon microscopy. Immunology-based techniques detect and purify exosomes using gold coating with a specific antigen. Inverted fluorescent microscopy was used to detect target exosomes quantitatively utilizing fluorescent nanospheres as the label. We quantified the expression of CD63 surface protein markers on exosomes from conditioned culture media of breast cancer cells. The exosomes that targeted specific proteins with controls were statistically analyzed and compared to those that targeted non-specific proteins. Results from SEM showed that the exosomes were circular, between 30 and 150 nanometers in size. The porous gold substrates captured more exosomes than the nonporous substrates. Nitric acid treatments at different times resulted in a variety of pore sizes. Despite the increase in the size of the pores, the number of exosomes found in the porous gold substrate treated for 10 min nearly doubled compared to the one treated for 5 min. In this work, a fluorescence biosensor was developed to detect breast cancer exosomes using nanoporous gold substrates (NPG). Assay and model exosomes of specific breast cancer cells showed that exosomes exhibit diagnostic surface protein markers, reflecting the protein profile of their parent cells. Furthermore, the specific binding between the exosome surface antibodies and the targets identified the CD63 biomarkers on the exosome, suggesting these markers’ diagnostic potential. This study can accelerate exosome research in determining tumor-related exosomes and develop novel cancer diagnostic methods.

## 1. Introduction

Exosomes serve as diagnostic tools for many diseases due to their function as cargo carriers of cellular components. Exosomes and their cargos are involved in the pathogenesis of numerous cancers, such as breast cancer [1,2]. Exosomes presently are under attention as biomarkers for breast cancer. There are three breast cancer cell lines (MCF7, MDA-MB-231, and SKBr3) [3]. As described by Jabbari et al., breast cancer exosomes such as T47DA18, MDA-MB-231, and MCF-7 could reach human primary mammary epithelial cells (HMECs) and induce reactive oxygen species formation, autophagy flow, and production of tumor factors from HMECs [4,5]. Exosomes are usually released into the extracellular space and enter circulation. These exosomes can be isolated from the blood, and their content reflects the parent breast cancer cells, enabling researchers to identify the breast tumor and its subtype [6]. Recent studies reported that the concentration of exosomes in breast cancer patients in serum or plasma ranges from $5.27\times 10^{9}\;\mathrm{to}\;1.31\times 10^{9}$ exosomes/mL [7]. Exosomes are different from traditional biopsies; exosomes rely on subcellular particles and their cargos [8].

Despite their diagnostic potential, the clinical use of exosomes as cancer biomarkers is still limited. One of the significant molecular detection and analysis challenges is their small size and complex biological environment. Exosomes need to be isolated and purified from cell culture supernatant or plasma before using them to ensure analytical precision. Traditional methods for exosome isolation are differential centrifugation, filtration, immunomagnetic separations, and microfluidics [9,10]. After purification, exosomes are usually analyzed for protein compositions using Western blot, enzyme-linked immunosorbent assays (ELISA), and mass spectrometry [11]. These standard approaches have helped understand exosome biology, but they are impractical for clinical use and longitudinal studies, consuming time, and labor. Exosome detection technologies for analysis have been greatly advanced in past years. Exosomes are detected based on fluorescence, surface plasmon resonance (SPR), light scattering plasmon resonance, nuclear magnetic resonance, electrochemical, and mechanical approaches [12]. Optical methods have been developed to enhance exosome quantification efficiency and multiplex detection. For example, fluorescence microscopy is used for in vitro imaging and for tracking exosomes labeled by specific fluorophores [13].

The innovation in this study is in combining nanotechnology with biomarkers and spectroscopy techniques. The research discussed the fabrication of nanoporous gold substrates (NPG) and the functionalization of the substrates. We discussed the mechanism to immobilize the biomarkers (MCF-7 exosomes), how to identify the peaks from the biomarker, and how to differentiate between different treatment times. We report the use of fluorescence for exosome detection and analysis. Using nanoporous gold substrate (NPG) can provide highly sensitive and specific detection by controlling the substrate’s surface chemistry, size, and structure [14,15]. Gold is well-known among various metal nanostructures as one of the most biocompatible plasmonic materials due to the properties of the collective oscillating modes of hot electron generation at the metal/dielectric interfaces [16]. NPG significantly enhances fluorescence, resulting in increased sensitivity [17]. The plasmonic properties of NPG substrates have been used for enhancements of fluorescence emission. NPG typically contains a network of voids that occupy over 70% of the substrate volume [16,18]. The immobilization of exosomes derived from the cancerous cell on nanoporous gold substrates (NPG) can provide a predictive biomarker for early cancer detection. The covalent immobilization on top of the nanoporous gold substrates was performed using a carboxylic group that conjugates CD63 antibodies with the gold surface for capturing MCF-7 exosomes. As such, a biosensor has a strong potential to discriminate MCF-7 exosomes from different biomolecules.

## 2. Materials and Methods

### 2.1. Materials

Silicon substrates, Ethanol, Nitrogen, Gold Source (AU), Silver Source (Ag), Chromium Source (Cr), Nitric Acid, *N*-(3-Dimethylaminopropyl)-*N*′-ethylcarbodiimide hydrochloride (EDC), *N*-Hydroxysuccinimide (NHS), Phosphate Buffer Saline (PBS), Protein G, Fluorescein (FITC), Phosphate-buffered Saline (PBS), PBST (Phosphate-buffered saline with Tween detergent), CD63 antibody (Biotin) # ab 134,331 (1 mg/mL), PBST, Exosomes (cell culture supernatant MCF7), and Vybrant Dill cell-labeling solution, ref: V22885).

### 2.2. Fabrication of Nanoporous Gold (NPG) Substrate

Gold nonporous (NPG) is generally made from a binary alloy such as gold (Au) and Silver (Ag) (Au-Ag) within a compositional range of 60–80% of Ag [14]. Dealloying is the process of removing the less noble element from a solid solution or intermetallic compound (e.g., Au-Ag, Au-Cu1, or Au-Al) using selective corrosion such as nitric acid (Figure 1). NPG is a promising candidate for the fabrication of biosensors. Its advantages include high surface area, mechanical and chemical stabilities, appropriate catalytic, biocompatibility, and tunable pore sizes [15,19].

Silicon substrates (crystal silicon wafers) were cut into square parts, cleaned with Ethanol then Nitrogen gas, and then the samples in the chamber of the sputtering machine of gold and silver targets (99.99% in purity) were added (Appendix A). The distance the substrate targets between was ≈10 cm. The pressure was fixed at 10 mTorr. The sputtering machine was switched on, and we waited for the pressure to reach 10.2 mTorr. Then, we started cleaning the chamber with Argon 10 SCCM for 10 min. The rotation speed was 30 rpm, and no intentional heating was applied to the substrate. After that, the Chromium source (100 W) (99.99% in purity) was opened for 12 min to create the Chromium adhesion layer to ensure proper adhesion of the gold/silver alloy films to prevent delamination of the gold when using the Nitric acid. Gold protection layer (50 W) was added for 3 min. The chamber was cleaned for 5 min with Ar (100 W). Electrical power applied to the gold (Au) and silver (Ag) targets were fixed to 25 and 100 W, respectively. Both gold and silver were added for 15 min; this yielded Au/Ag films with 75 atom% and 25 atom% silver and gold, respectively. As reported in the literature [20], an Ag concentration of 75% was chosen because minor differences were noticed when different concentrations were tested. Lastly, the resulting gold substrates were de alloyed in nitric acid for 5 min to form the nano porosity. The vacancies or the network voids in the alloy from three-dimensional nanoporous structures occupied over 70% of the substrate volume.

### 2.3. Exosomes Isolation and Characterization in Culture Media

MCF-7 cell lines were gifted from Dr. Shahab Khan’s Lab in cancer biology at HMC-Translation Research Institute (TRI). MCF-7 breast cancer cell lines were grown in DMEM cell culture media containing L-glutamine (2 mM), 10% Fetal bovine serum (FBS), penicillin, and streptomycin 100 U/mL (gibco by Life Technologies—15140-122). Cells were typically grown at 37 °C in a humidified atmosphere containing 5% CO_2_ incubator. Cell number and viability were measured using Trypan blue dye (0.4%) in the cell counter (TC 20 Automated Cell Counter). The MCF-7 cell lines were seeded at a concentration of 1 × 10^5^ cells/well, individually in 96-well plates in a volume of 200 μL cell culture media. After 24 h, the cells were permitted to grow and attach to the 96-well plates. After removing the supernatant, the exosome pellet was resuspended in cold, sterile PBS, and stored at −80 °C until use.

### 2.4. Integration of Exosomes with Nanoporous Gold (NPG) Substrate

Nanoporous gold substrates (NPG) were functionalized with exosomes using the covalent immobilization technique (Figure 2). The covalent immobilization of carboxylic acid (–COOH) groups on top of the nanoporous gold substrates was performed using reactive thiol-SH. Functionalizing gold surface with a carboxylic group helps conjugate the antibody or protein of interest with the gold surface for capturing exosomes [21]. As such, a biosensor has a strong potential to discriminate exosomes from different cellular origins for diagnostic implications. Figure 2 is a schematic overview of the integration of exosomes with NPG. A nanoporous gold substrate (NPG) was fabricated on a glass slide using a sputtering machine. Exosomes were recognized and immobilized on the nanoporous gold substrate (NPG) via the target-specific antibodies anchored on the surface of the substrate.

## 3. Results and Discussion

### 3.1. Characterization of Exosomes

The morphology of MCF-7 exosomes was investigated. The MCF-7 exosomes were characterized by scanning electron microscopy (SEM). As shown in Figure 3, the captured exosomes have a typical saucer-like structure with an average diameter of 152.8 nm, whose size was within the characteristic diameter range between 30 and 120 nm in diameter [22,23]. As revealed in the SEM image and compared to recent literature, the exosome’s results [20,21] and the isolated particle show typical saucer-like morphology of exosomes.

The exosome concertation was calculated by quantifying the exosome’s protein using a bicinchoninic acid (BCA) protein assay (Thermo Scientific, Waltham, MA, USA); each treatment was performed in triplicate (Table 1). The protein concentration calibration curve (Figure 4) was constructed with bovine serum albumin (BSA) as the standard. An excellent linear relationship (r = 0.9982) was obtained over the exosome’s concentration range. The samples were measured in a spectrophotometer at 530 nm. After calculation, the extraction of the protein concentration of exosomes was 2.09 μg·μL^−1^.

### 3.2. Characterization of Nanoporous Gold (NPG) Substrates

The resulting SEM images (Figure 5) show the difference in porous and nonporous gold substrates. Figure 5a shows the pores inside the porous gold substrate with an average pore size of ~174 nm, suitable for capturing exosomes ~(30 to 150 nm). Figure 5b shows the nonporous gold substrates with small pore sizes ~30 nm. Compared to previous work [15,22,23], the SEM results show similar nanostructures and morphology for porous and nonporous substrates.

#### 3.2.1. Functionalization of Nanoporous Gold (NPG) Substrate with Exosomes

The substrates were labeled by 4 μL of 2.09 μg/μL breast cancer exosomes, added to antibody-functionalized porous gold substrates, and incubated overnight on a shaker. Three groups were used in this experiment (Appendix A). Porous gold substrate functionalized CD63 antibody, porous gold substrate functionalized different antibodies (Control), and nonporous gold substrate functionalized with CD63 antibody [24,25,26,27]. Exosomes were detected with SEM and with inverted fluorescent microscopy. Fluorescence intensity was evaluated by using Annexin-V and PI for imaging.

As shown in the SEM images (Figure 6a), the exosome’s presence is a circular shape with a size between ~30 and 150 nm a mean size of ~152.8 nm. Compared to previous work [15,28,29], the exosomes are similar in shape and morphology. Porous gold substrates (Figure 6a) have more exosomes captured than the nonporous substrate (Figure 6b). The exosomes and different microvesicles were agglomerated in the nonporous substrate since the pore sizes were tiny compared to the porous gold substrates. Using the different antibodies (Figure 6c) in the control substrate, fewer other microvesicles were captured ~887.8 nm. MCF-7 exosomes can be captured using CD63 antibody as in Figure 6a.

Results from fluorescence analysis (Figure 7) show that exosomes can be seen using Fluorescein (594 nm). Moreover, dual-site affinity exists between the antibody and the exosomes. (Figure 7a) shows that the intensity of labeled exosomes using porous gold substrates is higher than the nonporous gold substrates (Figure 7b) and the porous gold substrates with exosomes only, without CD63 antibody (control) (Figure 7c). Figure 8 shows a comparison between the three fluorescence intensities as a percentage of reference (nonporous gold substrate). Fluorescence intensity in the porous gold substrates (Sample) was higher than the nonporous gold (Reference) due to the pore sizes. There was a larger area for the specific binding of exosomes and antibodies in the porous gold substrate. The porous gold substrates without antibodies (Control) had few non-specific bindings. The average number of exosomes was less than the porous gold substrates with antibodies.

#### 3.2.2. Functionalization of Nanoporous Gold (NPG) Substrate with Exosomes at Different Pore Sizes

The change of pore sizes of the porous gold substrates was achieved varying the nitric acid treatment times to obtain the different pore sizes (Table 2). Pore size was estimated as the average distance of the voids between the ligaments using several plan-view SEM micrographs recorded on each sample for different experimental groups. We were not able to evaluate precisely the vertical size of the pores due to the formation of ligaments under many dealloying conditions. However, we have remarked that when nanopores can be clearly identified, their vertical sizes are found to be large.

The same experiment was repeated for four samples treated with nitric acid for 5, 7, 10, and 18 min (Appendix A).

Fluorescence intensity was evaluated using Annexin-V and PI for imaging using inverted fluorescent microscopy. Figure 9 shows the fluorescence intensity for labeled exosomes at different pore sizes. Figure 9a shows that the fluorescence intensity using porous gold substrates for 5 min was 74,378.1 nm, which was less than the porous gold substrates treated for 7, 10, and 18 min (Figure 9b–d).

The average intensity (Figure 10) in the porous gold substrates treated for 10 min was almost doubled compared to the one treated for 5 min, due to the increase in the pore sizes. Gold substrates treated for 7 min and 10 min (Figure 10) have similar intensities, showing no huge difference in the pore sizes between the two samples. Figure 10 shows that the fluorescence intensity using porous gold substrates for 18 min was 126,244.2 nm. It is the highest fluorescence intensity; however, the exosomes were agglomerated inside the larger pores, as seen in Figure 10.

SEM results in Figure 11 show the average pore size for each porous gold substrate. As the treatment times increase, the pore sizes increase due to the de-alloy of silver from the nitric acid used in the experiment. Compared to a similar study, the pore sizes are proportionally increased by the treatment time [30,31]. We found that we can control the nanoporosity by controlling the treatment time. Nanoporous gold (NPG) is a promising candidate for developing biosensor devices [32]. According to Y. Xue et al., the de-alloying method has proven effective and successful in synthesizing three-dimensional nanoporous metals, such as Au, Ag, Pt, and Cu [26]. The ability to control the pore sizes in nanoporous gold substrates is an exciting advantage [32]. This feature can accommodate different targeted molecules and increase the captured biomolecules’ intensity such as exosomes. These exosomes can be easily detected using fluorescent labeling [33].

Figure 12 shows SEM results of the cross-section of the porous gold substrates at different magnifications. It is a sponge-like reticulate structure with nanometer-sized pores. The cross-section images clearly show the porosity of the gold substrates with large cracks from nanometer to microns.

Figure 13 shows the relation between the pore size and the fluorescence intensity. The highest intensity was found when the pore size was 330 nm. When the porous gold was treated for 18 min as the treatment time increased, the pore sizes increased. This is due to the morphological changes from the dealloying time. As the dealloying time increased, the pore size increased. Immobilization of the exosomes can be affected by the size and the shape of the nanopores. More exosomes can be immobilized in the pores. The fluorescence intensity is linearly related to the exosomes’ concentration. The performance of a porous structure depends on the distance between neighboring ligaments and the curvature of ligament [31]. Surface modification of porous substrates is crucial for triggering substantial local electromagnetic field enhancement around the roughened surfaces [18]. The tunable nanoporosity of the plasmonic NPG substrate aid in the fluorophore emissions at a peak of 594 nm, suggesting the presence of target exosomes.

## 4. Conclusions

The proposed device in this study has identified cancer biomolecules (exosomes) with blood-based biomarkers and nanotechnology. There was a dual-site affinity between the surface, the antibody, and the biomolecules. The functionalization layer on the porous and nonporous gold substrates worked as expected. The fabricated porous gold substrates can capture less than 150 nm biomolecules. The SEM results showed exosomes’ presence as a circular shape with a size between ~30 and 150 nm. Exosomes can be seen using Fluorescein at a wavelength 594 nm. There was a dual-site affinity between CD63 antibody and MCF-7 exosomes. Porous gold substrates have more exosomes captured compared to the nonporous substrate due to the pore sizes. There was a larger area for the specific binding of exosomes and antibodies in the porous gold substrate. Furthermore, fewer exosomes were captured when using the different antibodies in the control substrate, and when using porous substrates.

Also, when different treatment times of nitric acid were used. Different pore sizes were created. The fluorescence intensity using porous gold substrates for 5 min was 74,378.1 nm, which was less than the porous gold substrates treated for 7, 10, and 18 min. The average number of exosomes in the porous gold substrates treated for 10 min was almost doubled compared to the one treated for 5 min, due to the increase in the pore sizes. Gold substrates treated for 7 min and 10 min have similar intensities, which shows that there was no such huge difference in the pore sizes between the two samples. The fluorescence intensity using porous gold substrates for 18 min was 126,244.2 nm; this being the highest fluorescence intensity; however, the exosomes were agglomerated inside the larger pores.

## Figures and Tables

**Figure 1 sensors-22-06347-f001:**
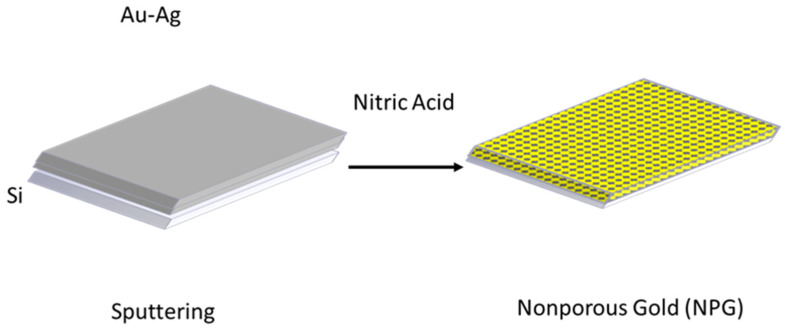
Nanoporous Gold (NPG).

**Figure 2 sensors-22-06347-f002:**
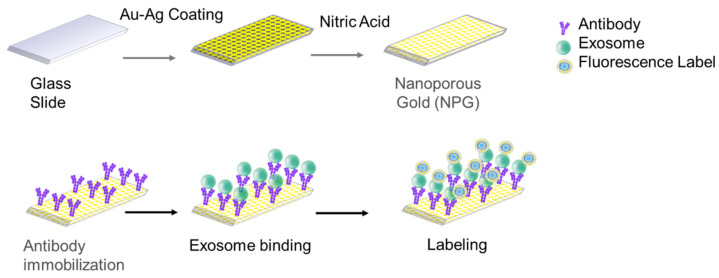
Schematic overview of the integration of exosomes with NPG. A nanoporous gold substrate (NPG) was fabricated on a glass slide using a sputtering machine. Exosomes were recognized and immobilized on the nanoporous gold substrate (NPG) via the target-specific antibodies anchored on the surface of the substrate.

**Figure 3 sensors-22-06347-f003:**
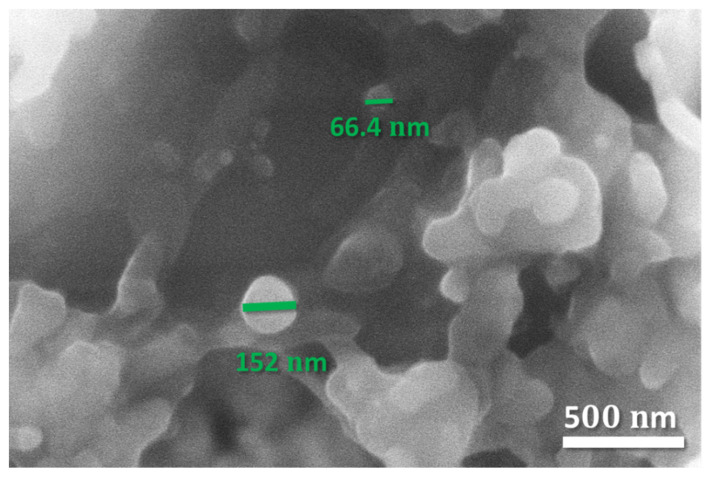
SEM (Scanning Electron Microscopy) image of MCF-7 exosomes on porous gold substate; the average size is 152 nm.

**Figure 4 sensors-22-06347-f004:**
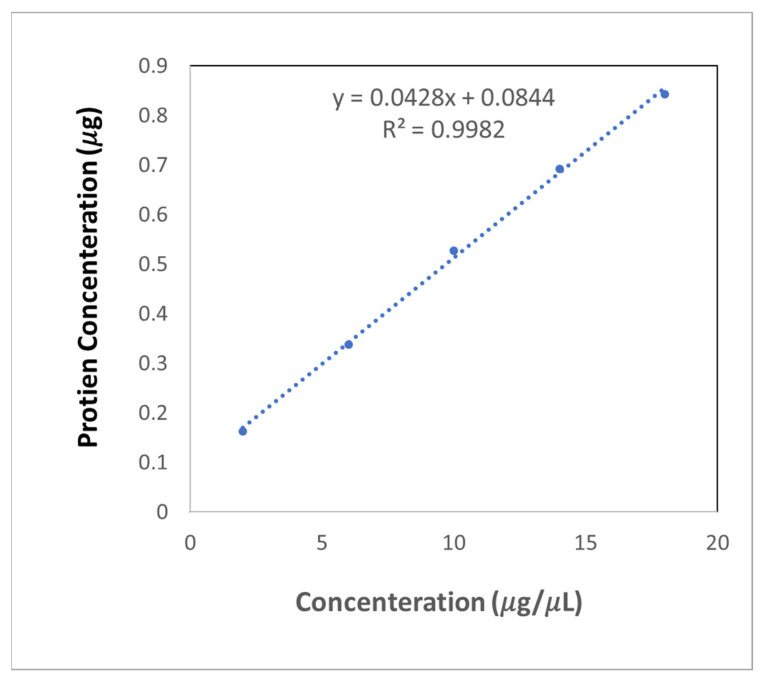
Exosomes Concentration Standard Curve.

**Figure 5 sensors-22-06347-f005:**
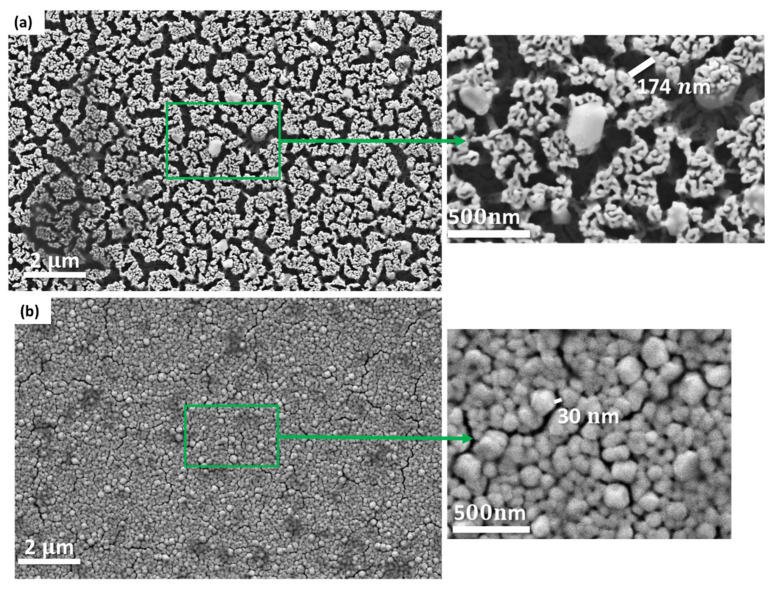
SEM images of the gold substrates: (**a**) porous gold substrate shows larger pore sizes 174 nm; (**b**) nonporous gold substrates with small pore sizes 30 nm.

**Figure 6 sensors-22-06347-f006:**
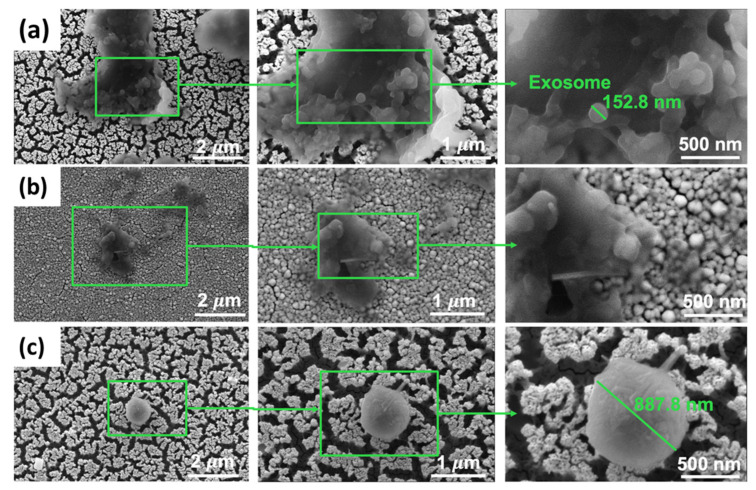
SEM Results of gold substrates functionalization: (**a**) Porous gold substrates functionalized with exosomes, (**b**) Nonporous gold substrates functionalized with exosomes (Reference), (**c**) Porous gold substrates functionalized with exosomes with different antibodies (Control).

**Figure 7 sensors-22-06347-f007:**
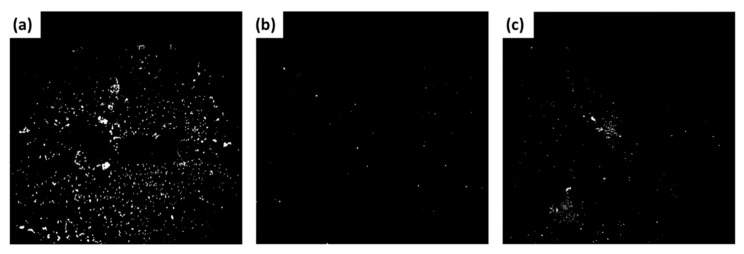
Fluorescence images of (**a**) porous gold substrates with antibody and exosomes; (**b**) porous gold substrate with exosomes only, without CD63 antibody (control); (**c**) nonporous gold substrate with antibody and exosomes (reference).

**Figure 8 sensors-22-06347-f008:**
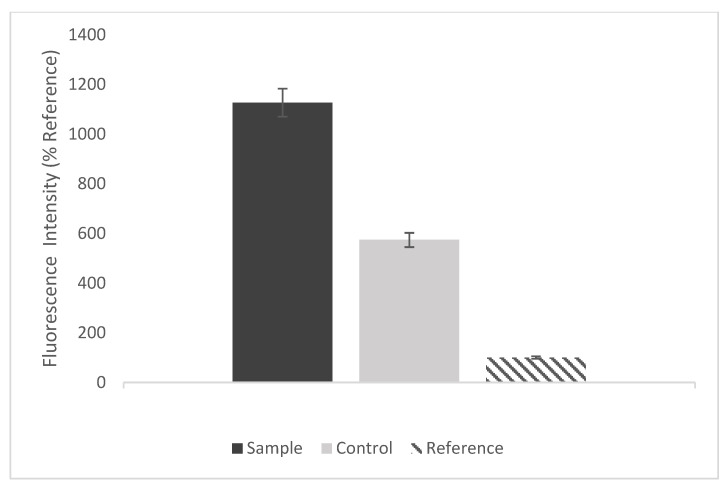
The fluorescence intensity as % of Reference.

**Figure 9 sensors-22-06347-f009:**
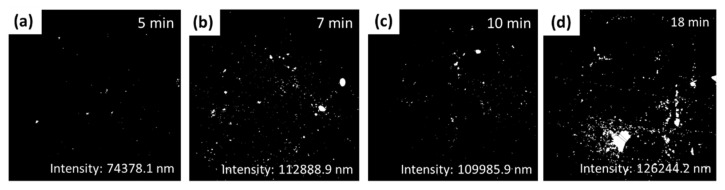
Fluorescence comparison between different treatment times: (**a**) Porous gold substrate treated for 5 min, average IntDen (Integrated Density) = 74,378.1 nm; (**b**) Porous gold substrate treated for 7 min, average IntDen (Integrated Density) = 112,888.9 nm; (**c**) Porous gold substrate treated for 10 min, average IntDen (Integrated Density) = 109,985.9 nm; (**d**) Porous gold substrate treated for 18 min, average IntDen (Integrated Density) = 126,244.2 nm.

**Figure 10 sensors-22-06347-f010:**
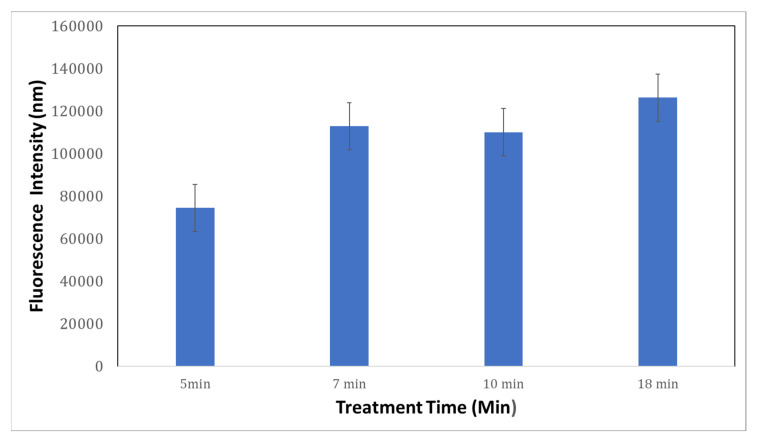
Comparison of average fluorescence intensity between different treatment times.

**Figure 11 sensors-22-06347-f011:**
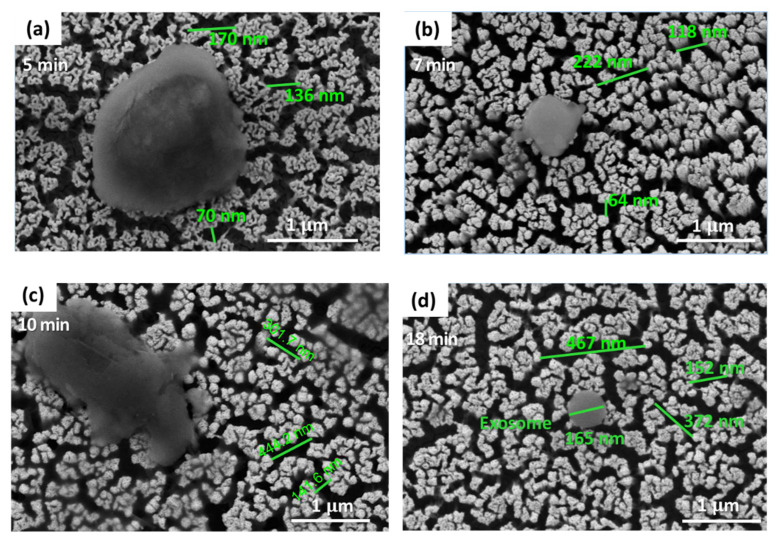
SEM micrograph showing porous gold substrates at different pore sizes: (**a**) Porous gold substrate treated for 5 min, (**b**) Porous gold substrate treated for 7 min, (**c**) Porous gold substrate treated for 10 min, (**d**) Porous gold substrate treated for 18 min.

**Figure 12 sensors-22-06347-f012:**
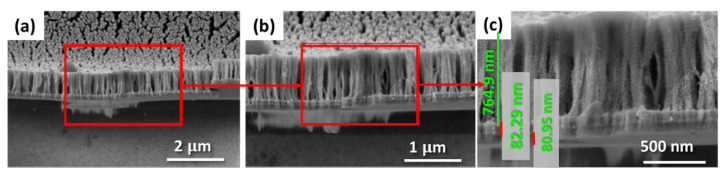
(**a**) Cross-section SEM micrograph of porous gold substrates at different magnification (**b**,**c**).

**Figure 13 sensors-22-06347-f013:**
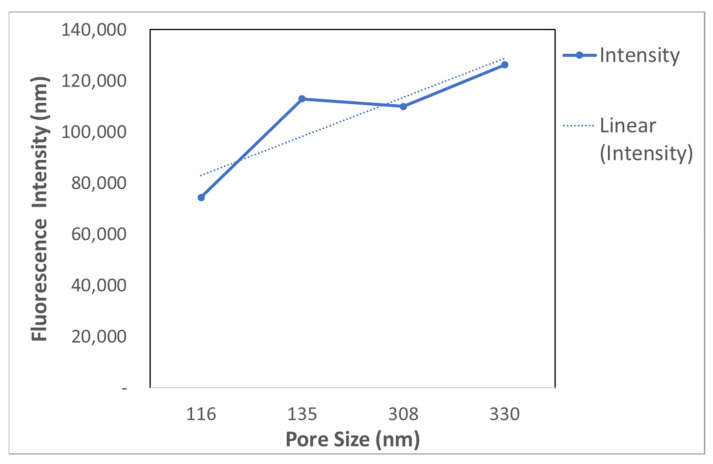
Pore Size vs. Fluorescence Intensity.

**Table 1 sensors-22-06347-t001:** Exosomes Concentration.

	Type	Concentration	1	2	3	Average
A	BSA0	0	0.085	0.085	0.094	0.088
B	BSA1	2	0.257	0.253	0.245	0.252
C	BSA3	6	0.438	0.417	0.422	0.426
D	BSA5	10	0.621	0.643	0.582	0.615
E	BSA7	14	0.770	0.775	0.796	0.781
F	BSA9	18	0.936	0.922	0.933	0.930
G	EXO1	2	0.253	0.280	0.269	0.267

**Table 2 sensors-22-06347-t002:** Different treatment times vs. different pore sizes.

Treatment Time	Average Pore Size (nm)
5 min	~116
7 min	~135
10 min	~308
18 min	~330

## Data Availability

Not applicable.

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
