# Peer review of "Quantification of Serum Exosome Biomarkers Using 3D Nanoporous Gold and Spectrophotometry"

_sensors, 2022, doi:10.3390/s22176347_

Round 1
Reviewer 1 Report
- Oddly enough, the author mentioned several times in the paper the preparation of an electrochemical sensor, but this work did not use such technology.
- The work involved assembling a very common immune sensor on a nano-substrate. I didn't find the work has enough innovation.
- It's hard for me to understand how such a random and uneven distribution of exosomes on a basis can produce stable results.
- The experimental results show that there is a positive correlation between pore size and fluorescence intensity, but there is no reasonable mechanism.
Based on the above concerns, I do not think this paper can be accepted for publication.
Author Response
Comments from the Editors and Reviewers:
Reviewer 1:
- Oddly enough, the author mentioned several times in the paper the preparation of an electrochemical sensor, but this work did not use such technology.
We thank the reviewer for his valuable comments and for pointing this out. The relevant electrochemical sensor has been removed from the revised manuscript.
2. The work involved assembling a very common immune sensor on a nano-substrate. I didn't find the work has enough innovation.
We appreciate the reviewer’s comment. We have added more details to the innovation part in the introduction. The innovation is in combining nanotechnology with biomarkers and spectroscopy techniques. The manuscript discussed how to fabricate the nanoporous gold substrate (NPG), the technique to functionalize the substrates, how to immobilize the biomolecules (MCF-7 exosomes), how to detect, how to identify the peaks from the biomarker, and how to differentiate between different treatment times.
3. It's hard for me to understand how such a random and uneven distribution of exosomes on a basis can produce stable results.
We thank the reviewer for his important comment. In the study, we were using inverted fluorescent microscopy to detect excitation of the fluorophore molecule as transduction components to exosomes (Due to the thiol group covalent interactions). In the resulted images the changes in the fluorescence were correlated to the initial conditions and to the non-porous substrates. The fluorescence intensity is detected and analyzed based on the amount of exosome bond. Unbound exosomes were washed, and we were able to differentiate the intensity of the fluorescence for different groups.
4. The experimental results show that there is a positive correlation between pore size and fluorescence intensity, but there is no reasonable mechanism.
We appreciate the reviewer’s comment. We were able to quantify the average fluorescence intensity using Annexin-V and PI for imaging using inverted fluorescent microscopy for different treatment times (different average pore sizes) at 5 min, 7 min, 10 min, and 18 min (Figure 10). We compared the average intensity for the four substrates used in the experiments. For example, 5 min results were the lowest compared to the other three due to the difference in the average pore sizes. Figure 13 shows the average pore size compared to the fluorescence intensity; the results clearly show a linear relation (positive correlation). This is due to the morphological changes from the dealloying time. As the dealloying time increased the pore size increased. Immobilization of the exosomes can be affected by the size and the shape of the nanopores. More exosomes can be immobilized in the pores. The fluorescence intensity is linearly related to the exosome’s concentration.
Reviewer 2 Report
In this paper the authors use a nanoporous gold substrate to link exosomes and detect them via fluorescence.
the assay is compared with similar measurements performed on standard bulk gold substrate. the porous substrate increases the intensity of fluorescence emission and enables a larger number of exosomes link to the substrate thanks to the surface morphology.
The topic is not new, porous gold has been extensively used to perform bioassays but the test with exosome could be interesting
anyway the paper is rather badly presented and there are a significant number of technical and conceptual errors that must be corrected before to consider it for publication.
- there are a large number of "text" errors along the manuscript. for example:
- line 87 "clean with Ag"...it should be Ar!;
- line 99, line 103: CO_2, 200 microliter;
- line 120-127: all completely wrong!
- line 172: "as in the SEM results"???? in english this has no meaning! probably "as shown from the SEM micrographs"
- line 186: "fluorescence results"???? also here. no meaning. "Results from fluorescence analyses...."
- captions of figure 10, 11 and 12 are wrong
- line 280: "intensity of exosome"???? the fluorescence intensity maybe,,,,,
- caption figure 4, different font style
2. it is not clear what Figure 3 shows, which substrate is used for this micrograph?
3. The introduction must be radically improved. this paper is on the use of NPG as substrate for fluorescence assay of exosome. there are tons of papers on similar topic and the state-of-the-art must be better discussed
4. the authors support their results considering the "pore size" on the NPG. this is the major critical error in the text. NPG pore size is NOT in the range reported here, but much smaller. the pore size mentioned in this paper is just the size of the "cracks" in the film obtained during not controlled de-alloying. this is a very well know effect in NPG preparation discussed in multiple papers.
5. The difference in fluorescence intensity is not discussed properly. NPG is a well known plasmonic material able to enhance emission of fluorofore in the range used here (560nm). different pore size modifies the enhanced in fluorescence emission and this is the real and correct justification of the observed effect. clearly different morphologies enable more or less efficient functionalization, also this just partially discussed.
6. I think that is a nonsense to report the fluorescence intensity in this way. It should be better to normalize the value with respect to 1, for example with respect to the value obtained with non-porous substrate. this enables to evaluate the enhancement factor
I think that this paper needs a radical revision and improvement. the authors should consider the state-of-the-art of NPG use in bioassay and starting from that trying to better discuss their (reasonably good) results
some useful papers to consider (to study and to include in the intro and discussion) are:
Phys. Chem. Chem. Phys., 2011,13, 3795-3799
Nanomaterials 2020, 10(4), 722
Nano Select 2021;2:1437–1458
ACS Nano 2021, 15, 4, 6038–6060
Author Response
Reviewer 2:
In this paper the authors use a nanoporous gold substrate to link exosomes and detect them via fluorescence.
the assay is compared with similar measurements performed on standard bulk gold substrate. the porous substrate increases the intensity of fluorescence emission and enables a larger number of exosomes link to the substrate thanks to the surface morphology.
The topic is not new, porous gold has been extensively used to perform bioassays but the test with exosome could be interesting
anyway the paper is rather badly presented and there are a significant number of technical and conceptual errors that must be corrected before to consider it for publication.
We thank the reviewer for his valuable comments and for pointing this out. Finding a noninvasive detection device for breast cancer is still a significant challenge that is not achieved yet. By using MCF-7 exosomes from breast cancer cells, we have tested the ability to detect those exosomes using a nonporous gold substrate. Peaks from the exosomes could be clearly identified using fluorescence emissions.
1. there are a large number of "text" errors along the manuscript. for example:
a. line 87 "clean with Ag"...it should be Ar!;
b. line 99, line 103: CO_2, 200 microliter;
c. line 120-127: all completely wrong!
d. line 172: "as in the SEM results"???? in english this has no meaning! probably "as shown from the SEM micrographs"
e. line 186: "fluorescence results"???? also here. no meaning. "Results from fluorescence analyses...."
f. captions of figure 10, 11 and 12 are wrong
g. line 280: "intensity of exosome"???? the fluorescence intensity maybe,,,,,
h. caption figure 4, different font style
We thank the reviewer for his important comment. The grammar, punctuation, spelling, usage, and consistency errors have been corrected.
2. it is not clear what Figure 3 shows, which substrate is used for this micrograph?
Corrected (Figure 3. SEM (Scanning Electron Microscopy) image of MCF-7 exosomes on porous gold substate, the average size is 152 nm)
3. The introduction must be radically improved. this paper is on the use of NPG as substrate for fluorescence assay of exosome. there are tons of papers on similar topic and the state-of-the-art must be better discussed
We would like to thank the reviewer for his important comments. The introduction has been updated, and the innovation in the research was added.
4. the authors support their results considering the "pore size" on the NPG. this is the major critical error in the text. NPG pore size is NOT in the range reported here, but much smaller. the pore size mentioned in this paper is just the size of the "cracks" in the film obtained during not controlled de-alloying. this is a very well know effect in NPG preparation discussed in multiple papers.
We appreciate the reviewer’s comment, and we agree with the comment about the pore size, we have updated the details on how we estimated the pore size (Line262: Pore size was estimated as the average distance of the voids between the ligaments using several plan-view SEM micrographs recorded on each sample for different experimental groups). We were not able to precisely evaluate the vertical size of the pores due to the formation of ligaments under many dealloying conditions. The fabrication of nanoporous gold substrates using the dealloying process does not require a homogeneous precursor alloy.
5. The difference in fluorescence intensity is not discussed properly. NPG is a well known plasmonic material able to enhance emission of fluorofore in the range used here (560nm). different pore size modifies the enhanced in fluorescence emission and this is the real and correct justification of the observed effect. clearly different morphologies enable more or less efficient functionalization, also this just partially discussed.
We totally agree with the reviewer's comment. We have added more details after Figure 12. “This is due to the morphological changes from the dealloying time. As the dealloying time increased the pore size increased. Immobilization of the exosomes can be affected by the size and the shape of the nanopores. More exosomes can be immobilized in the pores. The fluorescence intensity is linearly related to the exosomes concentration. The performance of a porous structure depends on the distance between neighboring ligaments and the the curvature of ligament [31]. Surface modification of porous substrates is crucial for triggering substantial local electromagnetic field enhancement around the roughened surfaces [18]. The tunable nanoporosity of the plasmonic NPG substrate aid with the fluorophore emission at peak 594 nm suggesting the presence of target exosomes”
6. I think that is a nonsense to report the fluorescence intensity in this way. It should be better to normalize the value with respect to 1, for example with respect to the value obtained with non-porous substrate. this enables to evaluate the enhancement factor
We appreciate the reviewer’s comment, and we agree that we should report the intensity with respect to the reference value. The figure was modified (Normalized using % of reference)
(Figure 8)
I think that this paper needs a radical revision and improvement. the authors should consider the state-of-the-art of NPG use in bioassay and starting from that trying to better discuss their (reasonably good) results
some useful papers to consider (to study and to include in the intro and discussion) are:
Phys. Chem. Chem. Phys., 2011,13, 3795-3799
Nanomaterials 2020, 10(4), 722
Nano Select 2021;2:1437–1458
ACS Nano 2021, 15, 4, 6038–6060.
We thank the reviewer for his valuable input to improve the quality of the manuscript and make it clearer. The relevant publications suggested by the reviewer have been cited in the revised manuscript.
Round 2
Reviewer 1 Report
The author has improved the quality of the paper, so I think this paper can be accepted for publication.
Reviewer 2 Report
The authors improved the paper according to my comments
I can now recommend the publication